# Combined Therapy of Low-Dose Angiotensin Receptor–Neprilysin Inhibitor and Sodium–Glucose Cotransporter-2 Inhibitor Prevents Doxorubicin-Induced Cardiac Dysfunction in Rodent Model with Minimal Adverse Effects

**DOI:** 10.3390/pharmaceutics14122629

**Published:** 2022-11-28

**Authors:** Donghyun Kim, Gyuho Jang, Jaetaek Hwang, Xiaofan Wei, Hyunsoo Kim, Jinbae Son, Sang-Jae Rhee, Kyeong-Ho Yun, Seok-Kyu Oh, Chang-Myung Oh, Raekil Park

**Affiliations:** 1Department of Biomedical Science and Engineering, Gwangju Institute of Science and Technology, Gwangju 61005, Republic of Korea; 2CNCure Biotech, Hwasun 58128, Republic of Korea; 3Department of Cardiovascular Medicine, Regional Cardiocerebrovascular Center, Wonkwang University Hospital, Iksan 54538, Republic of Korea

**Keywords:** angiotensin receptor–neprilysin inhibitor, sodium–glucose cotransporter 2 inhibitor, heart failure, doxorubicin, peroxisome proliferator-activated receptors

## Abstract

Although cancer-therapy-related cardiac dysfunction (CTRCD) is a critical issue in clinical practice, there is a glaring lack of evidence regarding cardiotoxicity management. To determine an effective and suitable dosage of treatment using angiotensin receptor–neprilysin inhibitors (ARNI) with sodium–glucose cotransporter 2 inhibitors (SGLT2i), we adopted a clinically relevant rodent model with doxorubicin, which would mimic cardiac dysfunction in CTRCD patients. After the oral administration of drugs (vehicle, SGLT2i, ARNI, Low-ARNI/SGLT2i, ARNI/SGLT2i), several physiologic parameters, including hemodynamic change, cardiac function, and histopathology, were evaluated. Bulk RNA-sequencing was performed to obtain insights into the molecular basis of a mouse heart response to Low-ARNI/SGLT2i treatment. For the first time, we report that the addition of low-dose ARNI with SGLT2i resulted in greater benefits than ARNI, SGLT2i alone or ARNI/SGLT2i combination in survival rate, cardiac function, hemodynamic change, and kidney function against doxorubicin-induced cardiotoxicity through peroxisome proliferator-activated receptor signaling pathway. Low-dose ARNI with SGLT2i combination treatment would be practically beneficial for improving cardiac functions against doxorubicin-induced heart failure with minimal adverse effects. Our findings suggest the Low-ARNI/SGLT2i combination as a feasible novel strategy in managing CTRCD patients.

## 1. Introduction

Cancer-therapy-related cardiac dysfunction (CTRCD) is a critically relevant issue in clinical practice; however, there is a glaring lack of evidence regarding cardiotoxicity management [1,2]. Clinical guidelines for CTRCD management recommend angiotensin-converting enzyme (ACE) inhibitors, angiotensin receptor blockers, and β-adrenoreceptor blockers as potential cardioprotective regimens [3,4,5]. However, angiotensin receptor–neprilysin inhibitor (ARNI) and sodium–glucose cotransporter 2 inhibitor (SGLT2i) have been included in the 2021 European Society of Cardiology guidelines for acute and chronic heart failure based on recent large-scale clinical trials [6,7,8].

ARNI, a combined drug of valsartan and sacubitril, is currently used for managing patients with heart failure [8]. However, the up-titration of ARNI is largely limited by symptomatic hypotension and can even prompt discontinuation of the drug [9]. Furthermore, we do not have sufficient evidence regarding the management of low blood pressure in concomitant medical therapy, such as ACE inhibitors, β-adrenoreceptor blockers, mineralocorticoid receptor antagonists, diuretics and SGLT2-inhibitors.

SGLT2i, a class of glucose-lowering agents, beneficially improves outcomes in patients with type II diabetes and a worsening heart failure event [10,11]. Quagliariello et al., reported that SGLT2i improves the myocardial strain by reducing fibrosis and pro-inflammatory cytokine levels in a rodent model of doxorubicin-induced cardiotoxicity [12]. Consistently, SGLT2i straighten up cardiac function by elevating the levels of ketone bodies in the same experimental model [13].

Although ARNI and SGLT2i have been proven to be the drugs of choice in treating heart failure [6,8], the underlying mechanisms of these two classes of drugs remain largely undefined. Efforts to fully define the molecular mechanism of the cardioprotective effect beyond the mode of action are ongoing [14,15,16]. Thus, investigating the interactions between the two drugs, which are known to have different cardioprotective mechanism, is required to develop a novel therapeutic strategy for CTRCD.

A recent retrospective study also showed that a combination of SGLT2i with ARNI was more effective in improving cardiac function and reducing the risks of hospitalization for heart failure and cardiovascular death in diabetic patients with heart failure with reduced ejection fraction compared with a single regimen of each drug [17]. However, it remains unclear whether the combined treatment with ARNI and SGLT2i exerts a synergistic or addictive effect on cardiac function of CTRCD.

Doxorubicin is frequently used in many types of cancers and is most widely-known for its association with CTRCD [18,19,20,21]. In this study, we adopted a rodent model with doxorubicin injection as a reliable preclinical model for patients with CTRCD to provide an effective and suitable dosage for ARNI with SGLT2i for improvement of cardiac dysfunction. Here, we demonstrated that the low-dose ARNI in addition to SGLT2i markedly improves doxorubicin-induced cardiac dysfunction with minimal adverse effects.

## 2. Materials and Methods

### 2.1. Animals

Male, 8-week-old, C57BL/6J littermate mice were purchased from Damul Science (Daejeon, Korea), housed in a temperature-controlled facility (22 ± 1 °C) that maintained a 12 h light/dark cycle, and provided free access to a standard chow diet (5L79, LabDiet, St. Louis, MO, USA) and water. A schematic representation of the study design is shown in Figure 1A. Mice were randomly assigned to each study group. To prepare the acute heart failure rodent model, mice were administered a single dose of 15 mg/kg doxorubicin (Sigma-Aldrich, Oakville, ON, Canada) or 0.9% sterile saline. To prepare the chronic heart failure model, serial intraperitoneal injections of doxorubicin (2.5 mg/kg) or 0.9% sterile saline were administered every 4 days for 24 days (cumulative dosage, 15 mg/kg). One day after doxorubicin injection, the control mice were gavaged daily with vehicle (corn oil) (Sigma-Aldrich), while the experimental mice were gavaged daily with ARNI (ENTRESTO^®^; Novartis, Basel, Switzerland) (68 mg/kg/day), SGLT2 inhibitor (Forxiga^®^; AstraZeneca, Cambridge, UK) (1 mg/kg/day), Low-ARNI/SGLT2i (Low-ARNI, 34 mg/kg/day; SGLT2i, 1 mg/kg/day), and ARNI/SGLT2i for 6 weeks. Dosage of ARNI and SGLT2i for our study was based on prior preclinical studies [22,23]. For the collection of blood and tissue samples, animals were euthanized by isoflurane overdose followed by exsanguination.

### 2.2. Tail-Cuff Plethysmography

Blood pressure was measured in conscious mice by tail-cuff plethysmography using a two-channel BP-2000 system (Visitech System, Apex, NC, USA). To minimize stress, each mouse was habituated to a blood pressure platform before the experiment began. Each session involved 20–30 measurements; the initial 10 cycles were acclimation cycles that were not included in the analysis.

### 2.3. Transthoracic Echocardiography

Echocardiograms were obtained using a Siemens Acuson NX3 ultrasound system (Siemens Healthineers, Erlangen, Germany) with 16-MHz linear array transducer. Mice were continuously monitored using a consistent isoflurane titration (1–2%) during imaging to maintain a heart rate of 450–500 beats per minute (bpm). M-mode (2-D guided) echocardiographic images were acquired along the parasternal short axis. Left ventricular (LV) internal dimensions in systole and diastole (LVIDs and LVIDd, respectively), as well as the diastolic thickness of the LV posterior wall (LVPWd) and diastolic intraventricular septum thickness (IVSd), were measured in at least three cardiac cycles and averaged. Fractional shortening (FS), ejection fraction (EF), and stroke volume were calculated as follows: FS (%) = (LVIDd − LVIDs) × 100/LVIDd; EF (%) = (LV end-systolic volume/LV end-diastolic volume) × 100; stroke volume = (LV end-diastolic volume − LV end-systolic volume).

### 2.4. Histopathological Analysis

Cross-sections of the mid-myocardium and kidney were fixed with buffer containing 10% formalin for 24 h and embedded in paraffin. Myocardial tissue sections were stained with hematoxylin and eosin (H&E), and kidney sections were stained with periodic acid–Schiff (PAS) reagent. To analyze the glomerular volume, 20–30 randomly selected superficial glomeruli from each kidney were measured, and calculations were performed using the Weibel–Gomez technique [24].

### 2.5. Protein Preparation and Western Blot Analysis

Protein concentration in heart tissue lysates was measured using the bicinchoninic acid method (Thermo Scientific, 23227, Waltham, MA, USA). Equivalent amounts of each protein extract were loaded into each well, separated by sodium dodecyl sulfate-polyacrylamide gel electrophoresis, and transferred to a nitrocellulose membrane. Blots were probed with the following antibodies: anti-βMHC (Abcam, Cambridge, UK, ab50967), anti-MyBPC3 (Santa Cruz Biotechnology, Dallas, TX, USA, sc-137237), anti-PPARα (Abcam, ab34509), anti-PPARβ/δ (Cell Signaling, Danvers, MA, USA, 2443S), anti-CD36 (Novus Biologicals, Minneapolis, MN, USA, NB400-144), anti-CPT1α (Abcam, ab1285568), anti-GLUT4 (Abcam, ab33780), anti-Akt (Cell Signaling, 9272S), anti-pS473 Akt (Cell Signaling, 9271S), anti-GSK3β (Cell Signaling, 5676S), anti-pSer9 GSK3β (Cell Signaling Technology, 9336S), and GAPDH (Proteintech, HRP-60004). Signals were visualized using the ChemiDoc Touch Imaging System (Bio-Rad, Hercules, CA, USA). Image Lab Software version 6.0.1 (Bio-Rad) was used to quantify the differences in the fold induction of protein expression normalized to that of GAPDH.

### 2.6. RNA Isolation and Real-Time qPCR Analysis

Total RNA was extracted from heart tissues using TRIzol^®^ reagent (Invitrogen Life Technologies, Carlsbad, CA, USA), and 2 μg of total RNA was reverse-transcribed onto complementary DNA (cDNA) with Transcriptor First Strand cDNA Synthesis Kit (Roche Life Sciences, Indianapolis, IN, USA). Quantitative real-time polymerase chain reaction (PCR) containing SYBR Green PCR Master Mix (Applied Biosystems, Austin, TX, USA) was performed using CFX Connect Real-Time PCR Detection System (Bio-Rad). PCR primers used in this study are listed in Appendix A.

### 2.7. Serum Biochemical Analysis

Blood samples were collected via cardiac puncture at the time of sacrifice. Plasma glucose levels were measured using a glucometer (Accu-Chek Active; Roche, Basel, Switzerland). Plasma levels of B-type natriuretic peptide (BNP; #EIAM-BNP-1, RayBiotech, Peachtree Corners, GA, USA), N-terminal pro-atrial natriuretic peptide (NT-proANP; #BI-20892, Biomedica, Vienna, Austria), and neutrophil gelatinase-associated lipocalin (NGAL) (#MLCN20, Minneapolis, MN, USA) were measured using ELISA kits according to the manufacturer’s instructions. Serum blood urea nitrogen (BUN; #K024-H1, Arbor assays, Ann Arbor, MI, USA), creatinine (#KB02-H1, Arbor assays, Ann Arbor, MI, USA), and β-hydroxybutyrate (#ab272541, Abcam) levels were measured using commercial kits.

### 2.8. RNA-Sequencing and Analysis of Transcriptome Data

Total RNA was extracted from frozen LV tissues using the RNeasy^®^ Mini Kit (Qiagen, Hilden, Germany). RNA samples (RNA integrity number > 8) were used for library preparation and sequencing. Poly(A)-enriched cDNA libraries were prepared using the Illumina TruSeq RNA Library Preparation kit. Library templates were then sequenced on NovaSeq 6000 sequencing system at a read length of 151 nt paired end, with a depth of approximately 50 million per sample (Illumina, San Diego, CA, USA). Four to five mice per drug-treated condition were used for subsequent data analysis.

### 2.9. Statistical Analysis

Quantitative data were analyzed using GraphPad Prism version 9.0 (San Diego, CA, USA). Between-group comparisons were performed using one-way or two-way analysis of variance followed by Tukey’s post hoc test for multiple comparisons based on the design of the experiment. Log-rank test was used to compare survival distributions. In the gene set enrichment analysis (GSEA), adjusted *p*-values corresponding to each normalized enrichment score (NES) were applied by Benjamini–Hochberg method. Statistical significance was set at *p* < 0.05.

## 3. Results

### 3.1. Combined Treatment with Low-ARNI/SGLT2i Prolongs the Overall Survival of Acute Doxorubicin-Injected Mice

To evaluate the beneficial effect of ARNI plus SGLT2i on chemotherapy-induced cardiomyopathy, we administered experimental drugs in rodent model of acute doxorubicin heart failure (Figure 1A). In pilot study for determining the appropriate combined dosages, we found that high doses of ARNI (136 mg/kg/day) aggravated the status of mice rather than protecting them. Drug down-titration is common for agents with a narrow therapeutic index to minimize the risk of adverse effects [25]. Based on the toxic side effects of high-doses of ARNI, we decided to add 50% of ARNI (34 mg/kg/day) to our experimental model.

In acute doxorubicin experiment, the drug-treated groups (ARNI, SGLT2i, Low-ARNI/SGLT2i, ARNI/SGLT2i treatment) showed improved survival rates compared with doxorubicin + vehicle group. Kaplan–Meier survival curves showed significant mortality (75%) in doxorubicin + vehicle group at experiment completion compared with that in the control mice. Additionally, the median survival time in doxorubicin + vehicle group was only 10 days (Figure 1B). In contrast, mortality in doxorubicin + Low-ARNI/SGLT2i group was significantly lower than that in doxorubicin + vehicle group (doxorubicin + Low-ARNI/SGLT2i: 14.2% vs. doxorubicin + vehicle: 75%, *p* < 0.05). Moreover, the survival rate of other groups, including SGLT2i only, ARNI only, and ARNI/SGLT2i combination, in acute doxorubicin-injected mice increased up to 66.7%, 50%, and 66.7%, respectively, compared with that of doxorubicin + vehicle group. However, Low-ARNI/SGLT2i group showed the highest survival (85.7%) compared with other three groups. Furthermore, an improvement in survival rate accompanied with weight gain in drug-treated groups (Appendix A). Treatment with doxorubicin + vehicle resulted in significant body weight loss (−23.8%) during the early course of study. Besides improved survival rate, treatment with Low-ARNI/SGLT2i markedly attenuated body weight loss (−15.9%) in doxorubicin-injected mice. Taken together, these results suggest that combined treatment with Low-ARNI/SGLT2i improves both survival rate and body weight in acute model of doxorubicin-induced heart failure.

Due to the anti-hypertensive effect of ARNI, blood pressure and pulse rate were measured to determine hemodynamic changes after ARNI and SGLT2i treatment in acute heart failure model. At 4 days after bolus injection of doxorubicin, systolic blood pressure of doxorubicin + vehicle group was lower than that of control mice (control: 114.8 ± 5.4 mmHg vs. doxorubicin + vehicle: 98.9 ± 5.1 mmHg, *p* < 0.05) (Figure 1D). Unexpectedly, treatment with ARNI showed dramatic dipping in systolic blood pressure to 66.7 ± 6.3 mmHg compared with other groups in acute model of doxorubicin-induced heart failure. Nonetheless, the dipping of blood pressure after combined treatment with SGLT2i, with either Low-ARNI (94.3 ± 9.3 mmHg) or ARNI (80.6 ± 14.5 mmHg), was alleviated in the presence of SGLT2i. Despite the hypotensive effect of ARNI in acute model of doxorubicin heart failure, the addition of ARNI along with SGLT2i significantly improved heart rate compared with that in the doxorubicin + vehicle group (control: 531 ± 37 bpm; doxorubicin + vehicle: 391 ± 44 bpm vs. doxorubicin + ARNI: 476 ± 27 bpm, *p* < 0.05; doxorubicin + vehicle vs. doxorubicin + Low-ARNI/SGLT2i: 482 ± 46 bpm, *p* < 0.05; doxorubicin + vehicle vs. doxorubicin + ARNI/SGLT2i: 487 ± 65 bpm, *p* < 0.05; Figure 1E). These observations indicate that combining of ARNI with SGLT2i help to maintain proper blood pressure and heart rate rather hypotension and bradycardia shown in acute doxorubicin-injected mice.

### 3.2. Combined Treatment with Low-ARNI/SGLT2i Recovers Doxorubicin-Induced Cardiac Dysfunction in Mice

We employed an established model of chronic cardiotoxicity to examine whether Low-ARNI/SGLT2i affects progressive cardiac dysfunction to mimic the clinical manifestations of CTRCD, as described in Figure 2A. We performed serial echocardiography to assess the cardiac function of mice before and after drug treatments. After doxorubicin injection, vehicle-treated mice had been developed progressive systolic dysfunction compared with control mice, evidenced by reduced left ventricular FS (control: 40.1 ± 1.0% vs. doxorubicin + vehicle: 29.8 ± 1.1%; *p* < 0.001) and left ventricular ejection fraction (LVEF) (control: 71.5 ± 1.2% vs. doxorubicin + vehicle: 58.1 ± 1.6%; *p* < 0.001) in a time-dependent manner (Figure 2C). In contrast, a single treatment with either ARNI or SGLT2i mitigated doxorubicin-induced reduction in LVEF (doxorubicin + ARNI: 67.5 ± 1.7%; doxorubicin + SGLT2i: 66.5 ± 0.5%). Additionally, combined treatment with Low-ARNI/SGLT2i alleviated LVEF reduction at 9 weeks in doxorubicin-injected mice (doxorubicin + Low-ARNI/SGLT2i: 73.8 ± 1.8%). Specifically, LVEF was comparable with baseline (baseline: 74.5 ± 1.6% vs. doxorubicin + Low-ARNI/SGLT2i: 73.8 ± 1.8%; *p* > 0.05) or the control group at 9 weeks (control: 71.5 ± 1.2% vs. doxorubicin + Low-ARNI/SGLT2i: 73.8 ± 1.8%; *p* > 0.05). Conversely, combined treatment with ARNI/SGLT2i in doxorubicin-injected mice failed to prevent this decline to baseline levels, whereas treatment with Low-ARNI/SGLT2i was effective in recovering FS and LVEF (FS, doxorubicin + Low-ARNI/SGLT2i: 41.5 ± 0.9% vs. doxorubicin + ARNI/SGLT2i: 36.4 ± 1.3%, *p* < 0.001; LVEF, doxorubicin + Low-ARNI/SGLT2i: 75.2 ± 0.8% vs. doxorubicin + ARNI/SGLT2i: 67.0 ± 1.7%, *p* < 0.001) (Figure 2C).

Furthermore, LV wall thinning after doxorubicin injection was significantly reduced in mice treated with either SGLT2i only or Low-ARNI/SGLT2i compared with that in doxorubicin + vehicle group (doxorubicin + vehicle: 0.99 ± 0.04 mm vs. doxorubicin + SGLT2i: 1.26 ± 0.08 mm, *p* < 0.001; doxorubicin + vehicle vs. doxorubicin + Low-ARNI/SGLT2i: 1.17 ± 0.05 mm, *p* < 0.01) (Figure 2D).

At baseline, there was no difference in LV end-diastolic volume (LVEDV), LV end-systolic volume (LVESV), and stroke volume in the control and experimental groups. After doxorubicin injection, stroke volume significantly declined compared with that in the control group (control: 44.3 ± 1.7 µL vs. doxorubicin + vehicle: 27.6 ± 1.8 µL, *p* < 0.001) (Figure 2E–G). Along with changes in systolic function, a time-dependent decline in stroke volume compared with baseline was observable after doxorubicin injection in the vehicle group (baseline: 45.1 ± 4.3 µL vs. 9 week: 27.6 ± 1.8 µL, *p* < 0.001), ARNI group (baseline: 44.8 ± 4.8 µL vs. 9 week: 33.0 ± 2.8 µL, *p* < 0.01), SGLT2i group (baseline: 45.2 ± 4.3 µL vs. 9 week: 36.5 ± 0.6 µL, *p* < 0.01), and ARNI/SGLT2i group (baseline: 45.4 ± 5.5 µL vs. 9 week: 38.6 ± 2.2 µL, *p* < 0.05). In contrast, after Low-ARNI/SGLT2i treatment, stroke volume recovered to a value comparable to that at the baseline (baseline: 44.8 ± 5.0 µL vs. 9 weeks: 41.6 ± 1.4 µL, *p* > 0.05; Figure 2G). Altogether, these results indicate that Low-ARNI/SGLT2i treatment significantly improves cardiac systolic function and LV structural remodeling in doxorubicin-injected mice. 

### 3.3. Combined Treatment with Low-ARNI/SGLT2i Attenuates Doxorubicin-Induced Cardiotoxicity

We next tested the effect of combined treatment on cardiac structure in doxorubicin-injected mice. There was a significant decrease in the ratio of heart weight and tibial length, indicating that doxorubicin injection caused progressive cardiac atrophy in the vehicle group. Conversely, treatment with Low-ARNI/SGLT2i exerted significant increases in the ratio of heart weight and tibial length as compared with doxorubicin + vehicle group (doxorubicin + vehicle: 56.5 ± 2.1 vs. doxorubicin + Low-ARNI/SGLT2i: 61.0 ± 2.1, *p* < 0.05) (Figure 3A,B).

To evaluate histopathological changes, we measured myofibrillar width and cytoplasmic vacuolization in heart section by H&E staining. As expected, doxorubicin + vehicle group exhibited the most severe abundance of cytoplasmic vacuoles (control: 4.8 ± 1.3 vs. doxorubicin + vehicle: 52.7 ± 5.7, *p* < 0.001) and thin myofibrils (control: 15.4 ± 1.3 µm vs. doxorubicin + vehicle: 9.0 ± 1.0 µm, *p* < 0.001) compared with the control group (Figure 2C,D). Both massive vacuolization and myofibrillar thinning shown in doxorubicin-injected mice decreased after treatment with SGLT2i only, ARNI only, and combination of two drugs (Figure 2C,E). Notably, the effects of Low-ARNI/SGLT2i on cytoplasmic vacuole and myofibrillar thinning were more pronounced than those of a single treatment with either ARNI or SGLT2i in doxorubicin-injected mice (doxorubicin + Low-ARNI/SGLT2i myofibril thickness: 13.9 ± 1.1 µm, *p* < 0.001; doxorubicin + Low-ARNI/SGLT2i cytoplasmic vacuoles: 13.3 ± 1.8, *p* < 0.001). 

We further measured the expression of cardiac structural proteins, including β-myosin heavy chain (βMHC) and myosin-binding protein C3 (MyBPC3). As shown in Figure 3G, a decrease in expression of βMHC and MyBPC3 after doxorubicin was recovered by treatment with Low-ARNI/SGLT2i. These data suggest that combined treatment with Low-ARNI/SGLT2i efficiently attenuates the pathological remodeling of cardiac structure in rodent model of doxorubicin-induced heart failure.

### 3.4. Combined Treatment with Low-ARNI/SGLT2i Stimulates the Secretion of Heart-Driven Hormones to Coordinate Kidney Function

To assess the pharmacological effect of ARNI, we measured the plasma levels of NT-proANP and BNP. As shown in Figure 4A,B, treatment with ARNI increased the circulating level of BNP, whereas it decreased NT-proANP levels compared with those in doxorubicin + vehicle group (NT-proANP: 2.3 ± 0.3 nmol/L vs. 1.3 ± 0.3 nmol/L, *p* < 0.001; BNP: 8.2 ± 2.4 pg/mL vs. 11.9 ± 2.4 pg/mL, *p* = 0.16). Among the other groups, Low-ARNI/SGLT2i group showed the highest value of BNP (doxorubicin + vehicle: 8.2 ± 2.4 pg/mL vs. doxorubicin + Low-ARNI/SGLT2i: 13.0 ± 1.8 pg/mL, *p* < 0.05; doxorubicin + ARNI: 11.9 ± 2.4 pg/mL, doxorubicin + SGLT2i: 10.5 ± 2.0 pg/mL, doxorubicin + ARNI/SGLT2i: 11.1 ± 5.4 pg/mL). Additionally, cardiac gene expression of ANP and BNP were decreased in the Low-ARNI/SGLT2i group (Figure 4C).

Next, we assessed the PAS staining and biomarkers of kidney injury. Kidney sections from control mice exhibited normal structure and size in PAS staining. In contrast, prominent damages in both glomerular and tubulointerstitial regions were observed in doxorubicin-injected mice (control: 19.5 ± 8.2 × 10^4^ µm^3^ vs. doxorubicin + vehicle: 12.6 ± 6.3 × 10^4^ µm^3^, *p* < 0.001). Administration of either ARNI or SGLT2i significantly ameliorated the shrinkage of glomerular volume (doxorubicin + ARNI: 14.9 ± 6.7 × 10^4^ µm^3^, doxorubicin + SGLT2i: 15.0 ± 6.4 × 10^4^ µm^3^). Consistently, the best recovery of glomerular shrinkage was shown in the Low-ARNI/SGLT2i group among all experimental groups (doxorubicin + vehicle: 12.6 ± 6.3 × 10^4^ µm^3^, doxorubicin + Low-ARNI/SGLT2i: 17.1 ± 7.1 × 10^4^ µm^3^, *p* < 0.001) (Figure 4E,F). A tendency of decline in serum BUN (17.4 ± 1.0 vs. 15.8 ± 0.9, *p* = 0.37) and creatinine levels (0.52 ± 0.12 vs. 0.50 ± 0.09, *p* = 0.9) was observed in the Low-ARNI/SGLT2i group, which was not statistically significant compared with that of doxorubicin + vehicle group (Figure 4G,H). Then, we measured serum NGAL, which is an early renal biomarker rapidly secreted by tubular cells in response to inflammation or nephron injury. A significant decline in serum NGAL level was observed in the Low-ARNI/SGLT2i group compared with that of doxorubicin + vehicle group (doxorubicin + vehicle: 127.8 ± 9.2 vs. doxorubicin + Low-ARNI/SGLT2i: 97.5 ± 11.0, *p* < 0.05) (Figure 4I). Collectively, our data imply that optimizing ARNI dosage in combined therapy would provide the most favorable protective effects through cardiorenal axis.

### 3.5. Effects of Combined Treatment with Low-ARNI/SGLT2i on Cardiac Energy Metabolism during Doxorubicin-Induced Cardiac Dysfunction

We performed bulk RNA-sequencing to obtain the mechanical insights of heart in response to ARNI and SGLT2i in chronic doxorubicin-injected mice. Principal component analysis of differentially expressed genes (DEGs) showed that the control, doxorubicin + vehicle, doxorubicin + ARNI, doxorubicin + SGLT2i, and doxorubicin + Low-ARNI/SGLT2i samples were clustered closely, indicating their similarity (Appendix A) [26,27]. Similarly, hierarchical clustering analysis also indicated that these samples were well grouped with each other (Appendix A) [28,29]. To understand the implications of global gene expression changes affected by combined treatment with Low-ARNI/SGLT2i, we performed GSEA [30]. GSEA showed that gene set was positively enriched with PPAR signaling pathway (NES 1.75, *p*.adjust = 0.008) and fatty acid degradation pathway (NES 1.67, *p*.adjust = 0.037) in doxorubicin + Low-ARNI/SGLT2i group compared with doxorubicin + vehicle group (Figure 5A). Genes in the k-means cluster 3 in hierarchical clustering, which are strongly upregulated in combined treatment with Low-ARNI/SGLT2i (228 genes), uncovered significant over-representation of DEGs in fatty acid metabolic process. The two dominant pathways were PPAR signaling (*Angptl4*, *Cpt1a*, *Cpt1b*, *Hmgcs2*, *Plin4*, *Scd4*, *and Slc27a1*) and fatty acid degradation (*Aldh9a1*, *Cpt1a*, and *Cpt1b*) in Kyoto Encyclopedia of Genes and Genomes pathway enrichment analysis (Figure 5B). Based on enrichment analysis, we examined the gene expression patterns of doxorubicin + Low-ARNI/SGLT2i groups according to Kyoto Encyclopedia of Genes and Genomes terms (“PPAR signaling pathway”) by comparing the control and doxorubicin + vehicle groups. Heat maps of relative mRNA expression of gene sets in PPAR signaling pathway are shown in Figure 5C. Fatty acid transport-related genes (*Slc27a1*, *Slc27a6*, *Acsbg1*, *Acsl3*, *Acsl4*, *Fabp4*, and *Fabp5*), fatty acid oxidation-related genes (*Cpt1a*, *Cpt1b*, *Acaa1a,* and *Ehhadh*), ketogenesis-related genes (*Hmgcs2*), and gluconeogenesis-related genes (*Pck1*, *Pck2,* and *Aqp7*) were upregulated in the Low-ARNI/SGLT2i group. Furthermore, recovery of dysregulated genes (*Rxrb*, *Acsl5*, *Dbi*, *Plin1*, *Acaa1a*, *Acox3*, *Sorbs1*, *Apoa2*, *and Slc27a4*) after doxorubicin injection was observable in doxorubicin + Low-ARNI/SGLT2i group. Thus, our transcriptome analysis suggests that combined treatment with Low-ARNI/SGLT2i improves metabolic flexibility in doxorubicin-injected heart through PPAR signaling pathway.

To further validate the functional relevance of identified genes associated with cardiac metabolism from RNA-sequencing data, we studied energy utilization in doxorubicin-induced cardiac injury. Initially, we examined renal glycosuria by measuring urine glucose levels of mice to assess the pharmacologic effects of SGLT2i. As expected, treatment with SGLT2i significantly increased glucose levels in urine (Figure 5D). Furthermore, to understand the resultant metabolic changes, we measured plasma glucose following drug administration. Treatment with doxorubicin resulted in a relatively low level of plasma glucose compared with the control (control: 244.7 ± 19 mg/dL vs. doxorubicin + vehicle: 190 ± 18.7 mg/dL, *p* < 0.001). Since SGLT2i is an anti-hyperglycemic agent that acts through glycosuria, it showed the lowest glucose levels among experimental groups. However, glucose level in SGLT2i group was not statistically significant compared with that of doxorubicin + vehicle group (doxorubicin + vehicle: 190 ± 18.7 mg/dL vs. doxorubicin + SGLT2i: 186 ± 20.6 mg/dL, *p* = 0.9). Interestingly, ARNI-treated group showed relatively higher levels of plasma glucose than doxorubicin + vehicle group (doxorubicin + ARNI: 209 ± 22.5 mg/dL, doxorubicin + Low-ARNI/SGLT2i: 222 ± 11.1 mg/dL, doxorubicin + ARNI/SGLT2i: 218 ± 20.2 mg/dL). However, only Low-ARNI/SGLT2i group was significantly different (*p* < 0.05) (Figure 5E). Since glucose is derived either exogenously or from glycogen stored intracellularly in heart, we assessed the level of glucose transporter type 4 (GLUT4) and glycogen synthase kinase 3β (GSK3β). GLUT4 expression was increased whereas inactivation of GSK3β through the protein kinase B (PKB/Akt) signaling pathway was up-regulated in the heart tissues from doxorubicin + Low-ARNI/SGLT2i group (Appendix A).

Moreover, the level of plasma β-hydroxybutyrate, an alternative fuel source in a failing heart, in both SGLT2i only and Low-ARNI/SGLT2i groups was significantly increased compared with that of doxorubicin + vehicle group (doxorubicin + vehicle: 245 ± 125 μM vs. doxorubicin + SGLT2i: 708 ± 206 μM, *p* < 0.01; doxorubicin + vehicle vs. doxorubicin + Low-ARNI/SGLT2i: 684 ± 180 μM, *p* < 0.05) (Figure 5F).

To ascertain whether the cardioprotective effect observed in the Low-ARNI/SGLT2i treatment was related to fatty acid metabolism, we evaluated the expression of proteins involved in fatty acid metabolism, including PPAR alpha (PPARα), fatty acid transporter (CD36), and carnitine palmitoyl transferase (CPT1α), from heart tissues. Immunoblot analysis showed that the expression of PPARα, CD36, and CPT1α was markedly increased in the Low-ARNI/SGLT2i group compared with doxorubicin + vehicle group (Figure 5G). Furthermore, quantitative RT-PCR revealed that the expression of genes, including *Cpt1a*, *Ehhadh*, *Acadl*, and *Hmgcs2*, involved in β-oxidation and ketogenesis was significantly upregulated after Low-ARNI/SGLT2i treatment (Figure 5H). Altogether, combined treatment with Low-ARNI/SGLT2i exerts protective effects on heart by modulating metabolic pathway, including glucose, ketone bodies, and fatty acids, in doxorubicin-induced model of heart failure.

## 4. Discussion

To the best of our knowledge, this study is the first observation to report the efficacy and suitable dosage of ARNI combined with SGLT2i in a preclinical model of CTRCD, and to show that low-dose ARNI in addition to SGLT2i exhibits a favorable effect on hemodynamic changes and enhanced myocardial metabolism through PPAR signaling pathway.

In our experiment, along with better survival, Low-ARNI/SGLT2i-treated group exhibited remarkable weight loss after doxorubicin injection (Appendix A). Early weight loss in patients with cancer negatively affects clinical prognosis [31]. Thus, we assumed that improved survival by Low-ARNI/SGLT2i may also be related to maintenance of weight and muscle mass. Although there is a limit in terms of inability to measure food and water intake, an increase in the plasma glucose levels of Low-ARNI/SGLT2i group may support this possibility. Since blood glucose levels vary according to dietary intake, Low-ARNI/SGLT2i groups may have taken sufficient nutrition to have increased blood glucose levels more than doxorubicin + vehicle group (Figure 5E).

In addition, reduced myocardial mass loss, augmented myofibril thickness, and an improvement of cardiac function by Low-ARNI/SGLT2i may be a result of either reduced degradation or enhanced synthesis of heart structural proteins, including βMHC and MyBPC3 (Figure 3G). In patients with CTRCD, cardiomyocytes are injured by the direct toxicity of chemotherapeutic reagents [32]. To overcome this issue, Low-ARNI/SGLT2i combination may support the possibility of regulating cardiac myocyte protein turnover for cardioprotection [33]. In this regard, it is important to elucidate the underlying molecular mechanisms through in-depth research. 

Heart failure shares many risk factors and numerous pathophysiological pathways with acute and chronic kidney disease through hemodynamic, neurohormonal, and cardiovascular disease-associated mechanisms [34]. The most well-established heart-derived hormones are ANP and BNP, which regulate the whole-body balance of water and electrolytes through target tissues [35]. As the kidney is a main target organ for natriuretic peptides, we tested PAS staining and serum NGAL level to assess kidney injury. In our study, the renal protective effect was obvious in combined Low-ARNI/SGLT2i treatment than in single treatment of either ARNI or SGLT2i (Figure 4E–I). In contrast to Low-ARNI/SGLT2i combination, ARNI/SGLT2i treatment induced marked renal impairment compared with others. Clinical trials such as Dapagliflozin and Prevention of Adverse Outcomes in Chronic Kidney Disease (DAPA-CKD), and Prospective Comparison of ARNI with ARB Global Outcomes in HF with Preserved Ejection Fraction (PARAGON-HF) reported that beneficial outcomes of ARNI and SGLT2i on kidney function. However, adverse event cases of acute kidney injury associated with SGLT2i and even ARNI/SGLT2i combination were also reported [36,37]. Major declines in kidney function under SGLT2i are often accompanied by acute illnesses such as diarrhea and/or sepsis [38]. Inappropriate ARNI doses induce a hemodynamically unstable condition and SGLT2i may lead to acute kidney injury by excessive intravascular volume depletion, due to diuresis and induction of renal medullary hypoxic injury [39]. Hence, we propose that combined administration of these two drugs in suitable doses would preserve organ function during chemotherapy through cardiorenal crosstalk. 

Due to the anti-hypertensive effect of ARNI, extreme blood pressure dipping with ARNI was observed in acute doxorubicin-induced model of cardiac dysfunction (Figure 1D). Although the dosage of ARNI was a human equivalent dose and comparable to that in previous preclinical studies, these results may need to be interpreted under highly virulent conditions induced by a single high-dose bolus injection of doxorubicin. Acute doxorubicin-induced model of heart failure mimics acute decompensated heart failure and similarly shows a vulnerable physiological state; therefore, even the modest amounts of anti-hypertensive agents can cause large drops. In Comparison of Sacubitril-Valsartan versus Enalapril on Effect on NT-proBNP in Patients Stabilized from an Acute Heart Failure Episode (PIONEER-HF) trial, ARNI group showed a higher tendency for symptomatic hypotension than ACE inhibitor group, although the difference was not statistically significant (ARNI: 15% vs. ACEi: 12.7%) [40]. These comparable rates of symptomatic hypotension may have critically affected the non-significant differences in secondary efficiency and safety outcomes in the PIONEER-HF trial. Therefore, ARNI dose modulation should be conducted with extreme caution in intra-hospital drug management of acute decompensated heart failure.

Furthermore, SGLT2i combined with ARNI reduced the risk of hypotension in acute doxorubicin model, implicating a clinical benefit for hemodynamically unstable patients with heart failure. Our result was consistent with the findings of Empagliflozin Outcome Trial in Patients with Chronic Heart Failure with Reduced Ejection Fraction (EMPEROR-Reduced), demonstrating a slight early increase in placebo-corrected systolic blood pressure among patients in the “<110 mmHg systolic blood pressure” group during the early phase of the trial [7]. In addition to the study by Böhm et al. [41], our study reassures the hemodynamic tolerability of SGLT2i, which seems to be a novel therapeutic intervention for managing low blood pressure.

The heart, an organ with a high metabolic rate, utilizes a wide range of substrates, including fatty acids, and glucose. During end-stage heart failure, the efficiency of fatty acid oxidation is significantly reduced [42]. To compensate for and increase the total energy supply, glucose and ketone bodies may be used for cardiac metabolism [43]. Transcriptome analysis employed in this study suggested that potential mechanisms in the Low-ARNI/SGLT2i group involve modulating cardiac metabolism through PPAR signaling pathway, mainly through fatty acid catabolism. PPARs are fatty acid sensors that regulate whole-body energy metabolism [44]. PPARα plays a crucial role in the body’s adaptive response to fasting by modulating fatty acid transport, oxidation, and ketogenesis [45]. PPARβ/δ activate both lipid and glucose metabolism, and are associated with body mass index, fasting glucose levels, and insulin resistance [46].

PPAR activation induces the expression of several protein-encoding genes involved in transmembrane transport and mitochondrial β-oxidation of fatty acids [47]. As shown in Figure 5C, Low-ARNI/SGLT2i treatment increased the expression of PPAR target genes involved in fatty acid transport (*Slc27a1*, *Slc27a6*, *Acsbg1*, *Acsl3*, *Acsl4*, *Fabp4*, *Fabp5*) and oxidation (*Cpt1a*, *Cpt1b*, *Acaa1a*, *Ehhadh*). Consistent with our transcriptomic data, immunoblotting and quantitative RT-PCR (Figure 5G,H) revealed that PPARα, CD36, and Cpt1a were increased in the Low-ARNI/SGLT2i group, which may contribute to the increased oxidation rates of myocardial fatty acid. This preservation of fatty acid oxidation suggests the promotion of catabolism for energy production, thereby improving myocardial energetics. Moreover, Low-ARNI/SGLT2i treatment induced myocardial expression of 3-hydroxy-3-methylglutaryl-CoA synthase 2 (Hmgcs2), a mitochondrial enzyme involved in the second rate-limiting step of ketogenic pathway. Ketone bodies are an alternative fuel source for failing hearts [48]. Our transcriptomic data of increased expression of *Hmgsc2* and elevation of circulating β-hydroxybutyrate in the Low-ARNI/SGLT2i treatment group indicate that ketones are also potential contributors to improving myocardial energetics (Figure 5F). Moreover, increased expression of GLUT4 and inactivation of GSK3β may influence the re-partitioning of glucose metabolism, which might have enhanced energy efficiency by generating higher ATP levels (Appendix A). Altogether, combined treatment with SGLT2i and suitable dosage of ARNI potentiates myocardial energetics through the utilization of fatty acid, glucose, and ketone bodies. 

We acknowledge the limitations of our study. Healthy animals were used in our study to yield the effect of ARNI and SGLT2i on doxorubicin-induced heart failure, but these drugs may interfere with chemotherapeutic efficacy. Furthermore, we could not measure diastolic function; therefore, it was not possible to evaluate whether diastolic function was changed by the combination of two drugs. Moreover, further investigations are needed to test the ability of ARNI and SGLT2i combination by metabolite profiling to identify the dynamic metabolic changes in the heart.

## 5. Conclusions

We report that the addition of low-dose ARNI with SGLT2i resulted in greater benefits than single treatment with either ARNI or SGLT2i in protecting cardiac function against doxorubicin-induced cardiotoxicity through PPAR signaling pathway. Our data may offer novel mechanistic insight into the benefits observed with Low-ARNI/SGLT2i combination. Even though future investigations are required to confirm the favorable role of ARNI and SGLT2i combined therapy in CTRCD, our findings would be applicable in developing optimized CTRCD management.

## Figures and Tables

**Figure 1 pharmaceutics-14-02629-f001:**
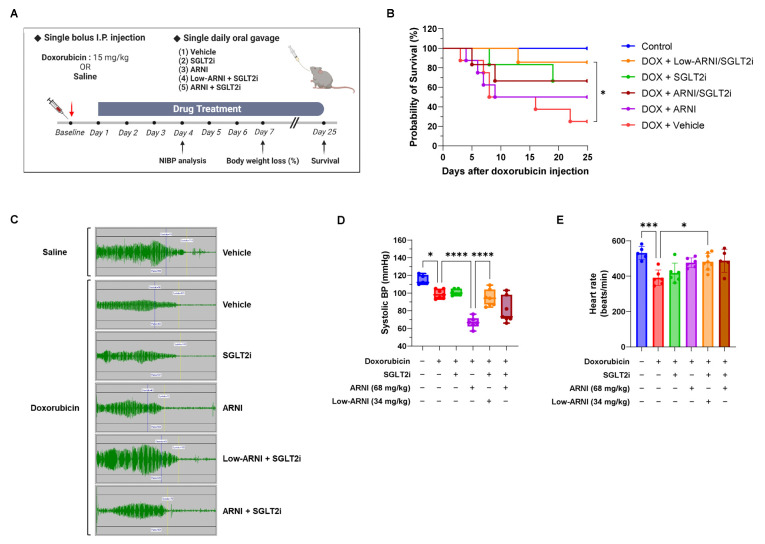
Survival gains and the hemodynamic effects of ARNI and SGLT2i treatment in acute doxorubicin-injected mice. (**A**) Schematic design of acute heart failure model (15 mg/kg i.p.). (**B**) Survival curves for 25 days after doxorubicin injection in indicated experimental groups (*n* = 6–8 per group). (**C**) Representative photoplethysmography at 4 days after doxorubicin injection. (**D**) Systolic blood pressure (*n* = 5–7 per group). (**E**) Heart rate at 4 days after doxorubicin injection (*n* = 5–7 per group). * *p* < 0.05, *** *p* < 0.001, ***** p* < 0.0001.

**Figure 2 pharmaceutics-14-02629-f002:**
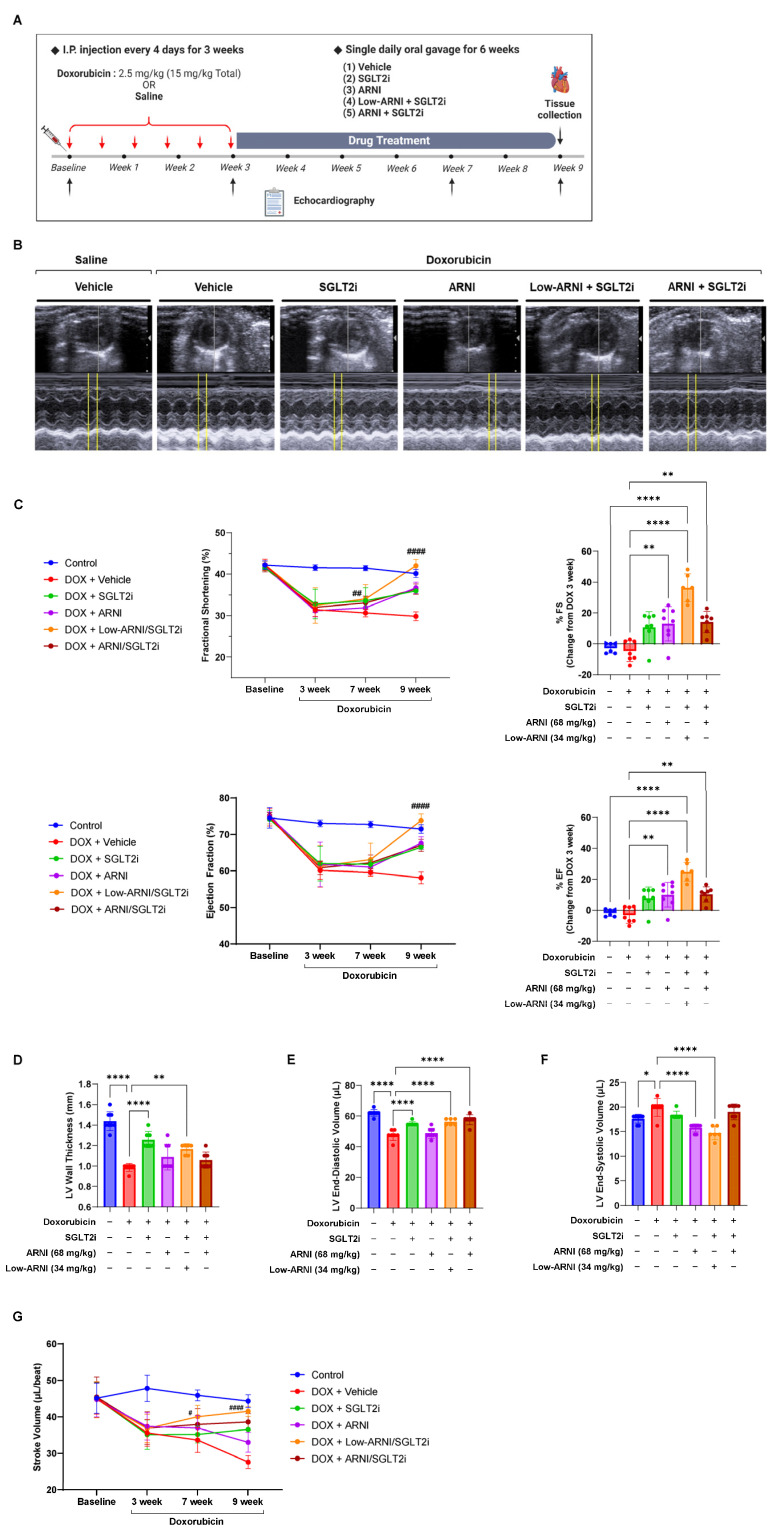
Recovery effects of Low-ARNI/SGLT2i treatment on doxorubicin-induced cardiac dysfunction. (**A**) Schema of chronic heart failure model (15 mg/kg total). (**B**) Representative M-mode echocardiography images after 9 weeks in indicated experimental groups. (**C**) Fractional shortening (FS%) and ejection fraction (EF%) at baseline and 3, 7, and 9 weeks after doxorubicin injection. (**D**) LV wall thickness. (**E**) LV end-diastolic volume. (**F**) LV end-systolic volume 9 weeks after doxorubicin injection. (**G**) Stroke volume at baseline and 3, 7, and 9 weeks after doxorubicin injection (*n* = 6–8/group). ^#^ *p* < 0.05, ^##^
*p* < 0.01, ^####^
*p* < 0.0001 compared to doxorubicin + vehicle group. * *p* < 0.05, ** *p* < 0.01, ***** p* < 0.0001.

**Figure 3 pharmaceutics-14-02629-f003:**
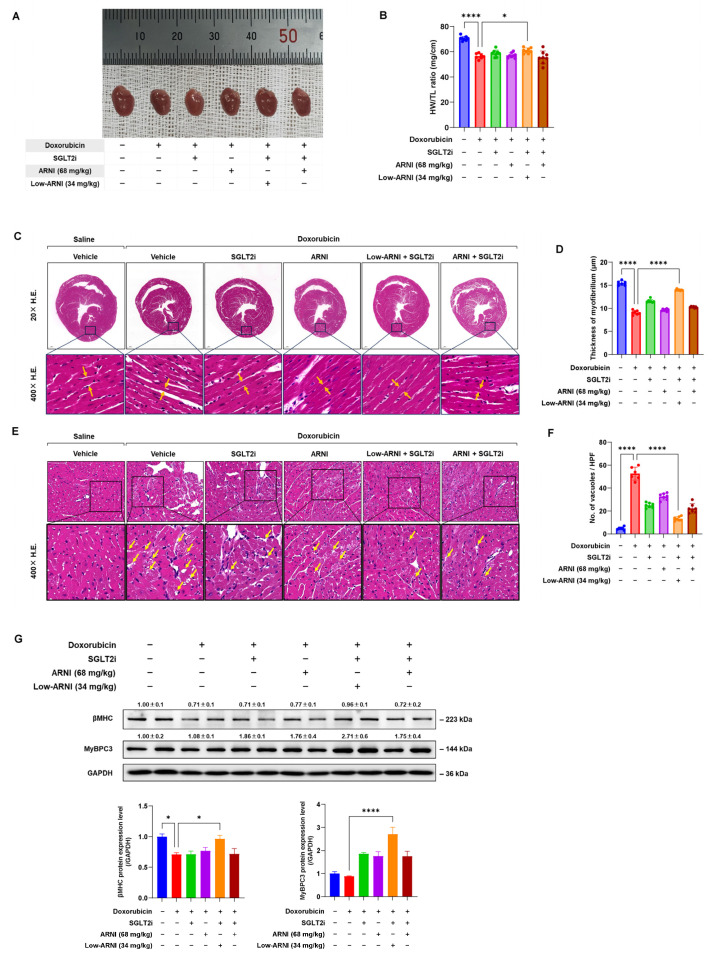
Protective effects of ARNI and SGLT2i treatment on doxorubicin-induced cardiotoxicity. (**A**) Whole-heart photograph. (**B**) The ratio of heart weight to tibia length (HW/TL) (*n* = 7–8/group). (**C**) Representative hematoxylin and eosin staining. (**D**) Quantitative myofibrillar thickness (*n* = 6–7/group) (**E**) Representative cytoplasmic vacuolization. (**F**) Quantitative number of cytoplasmic vacuoles (*n* = 6–7/group). (**G**) Representative immunoblots of βMHC, MyBPC3, and GAPDH and graphical quantification at 9 weeks after doxorubicin injection in indicated experimental groups (*n* = 4–5/group). * *p* < 0.05, **** *p* < 0.0001.

**Figure 4 pharmaceutics-14-02629-f004:**
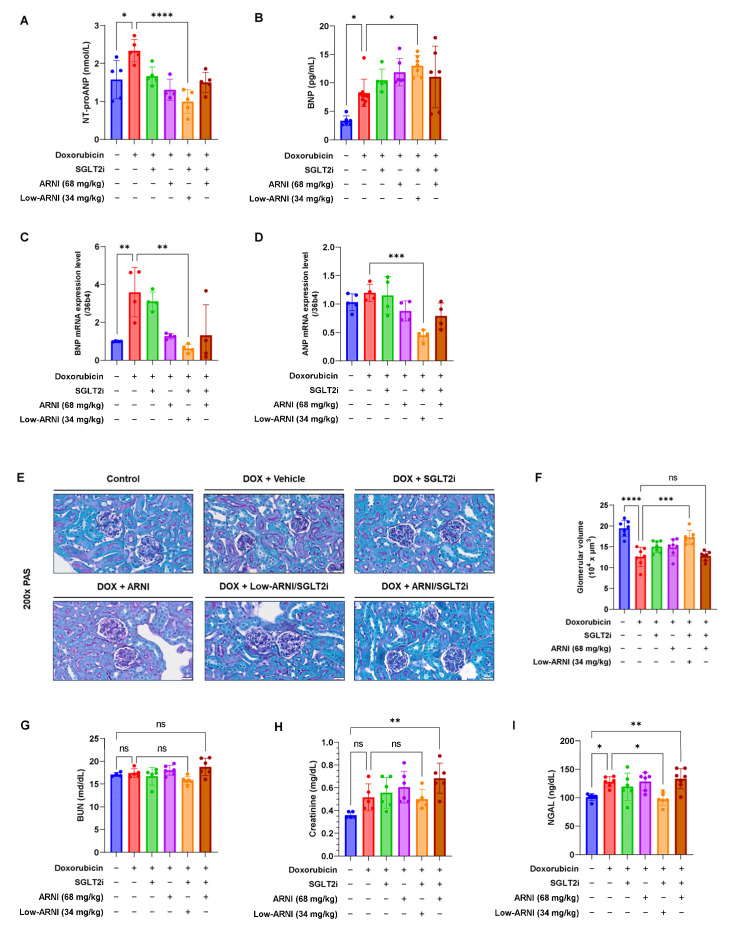
Effects of ARNI and SGLT2i treatment on heart-driven hormones in chronic doxorubicin-injected mice. (**A**) Plasma concentration of NT-proANP and (**B**) BNP 9 weeks after doxorubicin treatment initiation (*n* = 5–6/group). (**C**) mRNA expression of BNP and (**D**) ANP (*n* = 4–5/group). (**E**) Representative PAS staining of kidney cross-sections. (**F**) Quantitative mean glomerular volume (*n* = 6–7/group). (**G**) Plasma concentrations of BUN, (**H**) creatinine, and (**I**) NGAL 9 weeks after doxorubicin injection (*n* = 4–6/group). ns, no significance, * *p* < 0.05, ** *p* < 0.01, *** *p* < 0.001, ***** p* < 0.0001.

**Figure 5 pharmaceutics-14-02629-f005:**
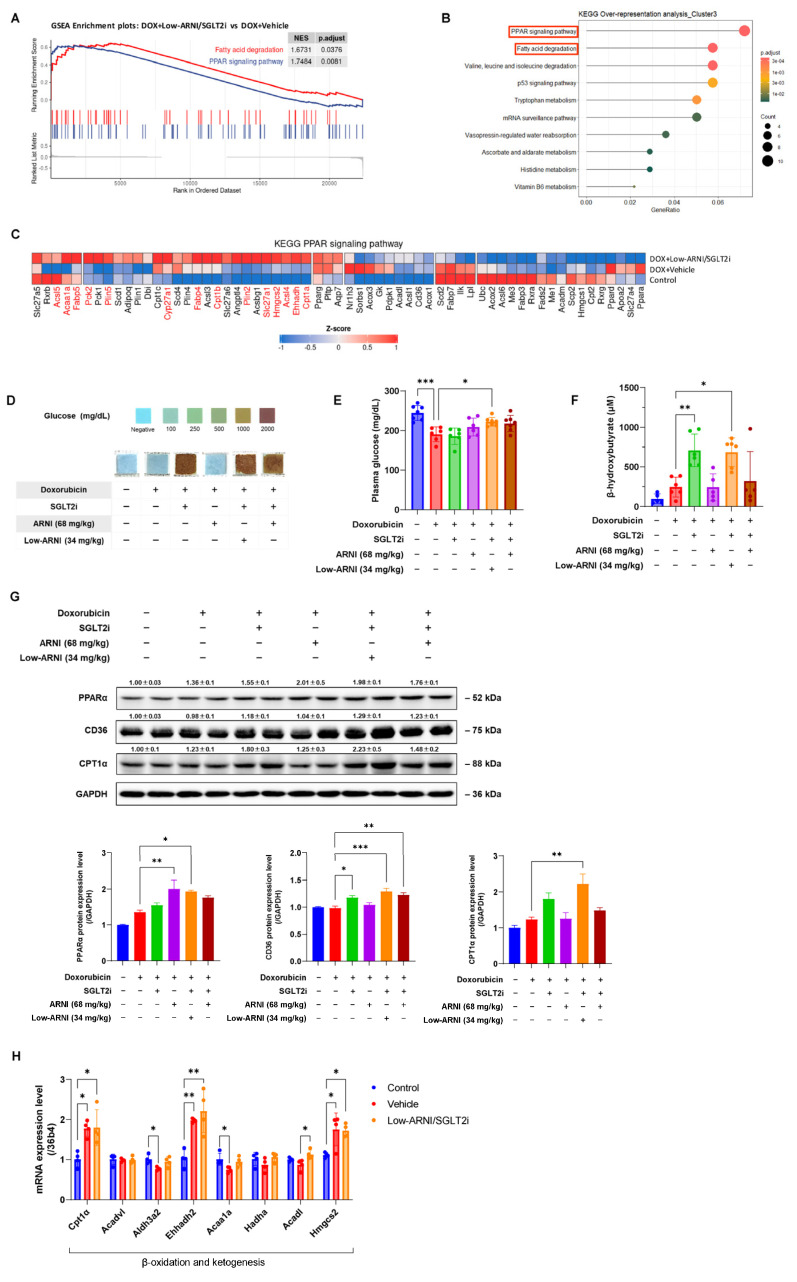
Effects of ARNI and SGLT2i treatment on cardiac energy metabolism in doxorubicin-induced cardiac injury. (**A**) GSEA plots associated with Low-ARNI/SGLT2i-treated groups. (**B**) Summary plot for over-representation analysis with Kyoto Encyclopedia of Genes and Genomes pathway of cluster 3 in hierarchical clustering. (**C**) Heat maps of relative mRNA expression of indicated genes in PPAR signaling pathway from control, doxorubicin + vehicle, and doxorubicin + Low-ARNI/SGLT2i groups. (**D**) Urine glucose, (**E**) plasma glucose and (**F**) plasma β-hydroxybutyrate levels (*n* = 5–7 per group). (**G**) Representative immunoblots of PPARα, CD36, and CPT1α protein at 9 weeks after doxorubicin injection in indicated experimental groups and graphical quantification (*n* = 4–5 per group) (**H**) mRNA levels of β-oxidation and ketogenesis-related genes are determined by qRT-PCR (*n* = 4 per group). * *p* < 0.05, ** *p* < 0.01, *** *p* < 0.001.

## Data Availability

The RNA-sequencing data associated with the current study are available at NCBI database under BioProject accession PRJNA890385, which can be accessed through the link http://www.ncbi.nlm.nih.gov/bioproject/890385 (accessed on 14 October 2022).

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
