# Peer review of "Combined Therapy of Low-Dose Angiotensin Receptor–Neprilysin Inhibitor and Sodium–Glucose Cotransporter-2 Inhibitor Prevents Doxorubicin-Induced Cardiac Dysfunction in Rodent Model with Minimal Adverse Effects"

_pharmaceutics, 2022, doi:10.3390/pharmaceutics14122629_

Round 1

Reviewer 1 Report

Kim et al. performed an investigation on effective and suitable dosages of treatment with angiotensin receptor-neprilysin inhibitors (ARNI) and sodium-glucose cotransporter 2 inhibitors (SGLY2i) using a relevant rodent model, which mimics cancer therapy-related cardiac dysfunction (CTRCD) in patients. The findings of this manuscript suggest that the combination of the low dose of ARNI with SGLT2i might lead to improvement of CTRCD conditions in patients and thus could be a novel treatment strategy.

Overall, this paper contains original results and might be interesting for specialists, which are developing therapies for CTRCD in patients.

I would recommend it for publication after the authors address my comments below:

1)    Line 87. What is the exact dosage of ARNI and SGLT2i for Low-ARNO/SGLT2i?

2)    Fig. 1C. Please improve the quality of the presented photoplethysmographs.

3)    Fig. 1D. Information about the number of mice per group is missing.

4)    Fig. 2. Please check if the descriptions in the figure legend are correct. Most likely the legend corresponds to Fig. 3.

5)    Fig. 3. Please check if the descriptions in the figure legend are correct. Most likely the legend corresponds to Fig. 2.

6)    Why 34 mg/kg was chosen as a low dosage of ARNI? Please explain it in the main text.

7)    Are different dosages of SGLT2i in combination with ARNI/Low-ARNI tested? If not please explain why.

Reviewer 2 Report

Authors have studied the effects of combined therapy to cardiac dysfunction. This study might draw reader’s attention. I would suggest to incorporate a few references regarding combined therapy of cancer remission namely, 10.1016/j.chaos.2021.110789; 10.1016/j.chaos.2020.109806. Although, authors have added discussion, I would also suggest enhancing the conclusion in details.

Reviewer 3 Report

Thank you for giving me for reviewing this interesting paper. However, the work was inappropriately designed and used doses were not selected properly. In addition, the protein expression turnover along the given treatment and treatment duration may not necessarily aligned. In additions, authors conclusion linked directly to human findings while these findings were specific to the used model and would need further work to be extrapolated and generalized. Therefore, this study findings should not be considered for publication in current status.     

Reviewer 4 Report

1. The study presents the results of original research.

2. Results reported have not been published elsewhere.

3. Experiments, statistics, and other analyses are performed to a high technical standard and are described in sufficient detail.

4. Conclusions are presented in an appropriate fashion and are supported by the data.

5. The article is presented in an intelligible fashion and is written in standard English.

6. The research meets all applicable standards for the ethics of experimentation and research integrity.

7. The article adheres to appropriate reporting guidelines and community standards for data availability.

Round 2

Reviewer 3 Report

All comments were addressed.